# Coverage and determinants of deworming uptake among under-five children in Somalia: A multilevel analysis of the 2020 SDHS data

Abdirahman Omer Ali[1,2], Awo Mohamed Kahie[1,3], Muhyadin Yusuf Dahir[3],
Suhaib Mohamed Kahie[2], Abdisalam Mahdi Hassan[1,2], Md. Moyazzem Hossain[4]*

1 Amoud University College of Health Science, School of Medicine, Borama, Somalia, 2 Office of Community Services, Amoud University, Borama, Somalia, 3 School of Postgraduate Studies and Research, Amoud University, Borama, Somalia, 4 Department of Statistics and Data Science, Jahangirnagar University, Savar, Dhaka, Bangladesh

* hossainmm@juniv.edu

## Abstract

### Background

Soil-transmitted helminth (STH) infections are a major public health concern in Somalia, particularly affecting the health and development of children under five. Therefore, this study aimed to assess the coverage of deworming uptake and identify associated multilevel factors with deworming uptake among Somali children aged 12–59 months using a Multilevel logistic regression model.

### Methods

This study analyzed data of 15,074 children aged 12–59 months from the 2020 Somalia Demographic and Health Survey (SDHS). Chi-square test and multilevel logistic regression were used to examine individual (maternal/child characteristics, health service use) and community (residence, region) factors associated with non-receipt of deworming medication (poor uptake).

### Results

Only 8.0% of children had received deworming medication, indicating critically low national coverage. The variations of poor deworming uptake among children of different ages in months were 92.91% between 12–15 months, 91.75% between 16–19 months, and 91.26% between 20–59 months. Poor deworming uptake was varied among maternal age groups, with rates of 92.10% (15–24 years), 91.89% (25–34 years), and 91.60% (35–49 years). Findings depict that significant regional variations existed. Better uptake was associated with higher maternal age and education, greater wealth, maternal employment, health facility delivery, and urban/nomadic residence (vs. rural). Residing in urban (AOR: 0.65; 95% CI: 0.51, 0.82, $p < 0.05$) or

**Data availability statement:** The data is freely accessible through the Somali National Data Archive (SoNADA) website at https://microdata.nbs.gov.so/index.php/catalog/50.

**Funding:** The author(s) received no specific funding for this work.

**Competing interests:** The authors have declared that no competing interests exist.

nomadic areas (AOR: 0.40; 95% CI: 0.32, 0.49, $p < 0.05$) was significantly associated with lower odds of poor uptake compared to rural areas. Unexpectedly, children without recent episodes of diarrhea had significantly higher odds of not receiving deworming treatment (AOR = 6.26).

## Conclusion

Low deworming coverage among under-5 children in Somalia is observed. Factors include higher maternal education, greater wealth, health facility delivery, urban or nomadic residence compared to rural, and older child age are significantly associated with deworming coverage. To improve the deworming coverage, organizing school-based campaigns and deploying mobile health teams for door-to-door visits in remote areas may be useful.

## Introduction

Soil-transmitted helminth (STH) infections, primarily caused by roundworm (Ascaris lumbricoides), whipworm (Trichuris trichiura), and hookworms (Necator americanus and Ancylostoma duodenale), represent a major global public health challenge, particularly affecting vulnerable populations in low- and middle-income countries (LMICs) [1]. Transmission occurs through ingestion or skin penetration of eggs or larvae present in contaminated soil, often linked to inadequate sanitation, poor hygiene practices, and lack of access to clean water [2]. Children under the age of five are especially susceptible due to their developing immune systems, frequent contact with soil during play, and poorer hygiene habits [3]. Chronic STH infections in this age group can lead to a cascade of detrimental health consequences, including malnutrition, iron-deficiency anemia, impaired cognitive development, and reduced physical fitness, thereby hindering their overall well-being and future potential [4,5] Deworming, the periodic administration of safe and effective anthelmintic drugs (like albendazole or mebendazole), is a cornerstone public health intervention recommended by the World Health Organization (WHO) to control STH morbidity through preventive chemotherapy (PC) programs targeting high-risk groups, including preschool-aged children [6]. For children under five, the critical window for growth and development, STH infections inflict damage that can last a lifetime, perpetuating cycles of poverty and poor health [7,8]. While effective and affordable treatments exist, delivering them to all eligible children—particularly in fragile settings like Somalia—remains a significant challenge [9]. Understanding the reach and determinants of deworming programs is not merely an academic exercise; it is fundamental to safeguarding the health and developmental trajectory of millions of children, unlocking their potential to thrive and contribute to their communities [4,10], and addressing this preventable condition is a crucial step towards achieving sustainable development goals related to child health and well-being [11].

The silent scourge of intestinal worms affects over a billion people worldwide, yet its burden falls most heavily on the youngest and most defenseless members

of society [12]. Globally, significant strides have been made in controlling STH infections through coordinated efforts spearheaded by the WHO, aiming to eliminate morbidity in children through regular PC campaigns [6,13]. Despite progress, Sub-Saharan Africa continues to bear the brunt of the global STH burden, with hundreds of millions estimated to be infected [14]. Factors such as widespread poverty, limited access to water, sanitation, and hygiene (WASH) facilities, climatic conditions favorable for parasite survival, and often overburdened health systems contribute to the persistence of high transmission rates across the continent [15,16]. Within the Horn of Africa, these challenges are often amplified by recurrent droughts, food insecurity, political instability, and large-scale population displacement, creating environments where parasitic infections flourish [17].

Somalia, situated in this volatile region, has historically faced complex humanitarian emergencies and possesses a fragmented health system, making its population, particularly young children, highly vulnerable to neglected tropical diseases like STH infections [18,19]. Somalia's specific context presents unique obstacles to effective public health interventions, including deworming programs for under-fives. Decades of conflict and instability have severely weakened health infrastructure and service delivery across large parts of the country [20]. Access to basic WASH facilities remains critically low, particularly in rural areas and among internally displaced populations, facilitating the transmission of STH [21,22]. While national deworming campaigns may be intended, their actual coverage and effectiveness are likely influenced by a complex interplay of factors, including security constraints, geographic accessibility, caregiver awareness and health-seeking behaviors, and the capacity of local health systems or implementing partners [19,23,24]. Poor deworming uptake among children aged 24–59 months was significantly associated with the mother's education, employment status, home birth, diarrhoea during the previous two weeks, and region of residency [25–27]. Several previous studies consider children aged 12–59 months for exploring the coverage and risk factors of deworming [26,28–30], which motivates the authors to include children aged 12–59 months in this study. Deworming may be an effective technique for preventing poor health outcomes in children [29]. Reliable, up-to-date information on the prevalence of deworming uptake and the factors influencing it among children under five in Somalia is scarce. The Somalia Demographic and Health Survey (SDHS) 2020 provides a crucial and timely opportunity to assess the situation using nationally representative data [31,32]. While previous studies might have touched upon child health indicators in Somalia, few, if any, have specifically focused on deworming in this age group using strong statistical methods applied to current national data. The primary novelty lies in the application of multilevel analysis. This approach allows for the simultaneous examination of both individual/household-level factors (e.g., maternal education, wealth quintile, child's age) and community-level or contextual factors (e.g., region, place of residence, potential proxies for local health service availability) associated with deworming status. Recognizing that factors influencing health behaviors operate at multiple levels, this methodology provides more nuanced and contextually relevant insights compared to traditional regression analyses that ignore the hierarchical structure of the data [33]. This study is among the first to use multilevel modeling on SDHS 2020 data to assess both individual- and community-level determinants of deworming uptake in Somali children. Unlike traditional analyses, this method accounts for data clustering and context-specific influences, offering more robust and actionable insights.

## Methods and materials

### Study design, setting, and data

This study employed a cross-sectional design using secondary data from the 2020 Somalia Demographic and Health Survey (SDHS).

### Sampling procedures

The SDHS followed a three-stage stratified cluster sample design in urban and rural strata with a probability proportional to size, for the sampling of Primary Sampling Units (PSU) and Secondary Sampling Units (SSU) (respectively at the first

and second stage), and systematic sampling of households at the third stage. For the nomadic stratum, a two-stage stratified cluster sample design was applied with a probability proportional to size for sampling of PSUs at the first stage and systematic sampling of households at the second stage. A total of 220 EAs and 150 EAs were allocated to urban and rural strata, respectively, while in the third stage, an average of 30 households were selected from the listed households in every EA to yield a total of 16,360 households that were eligible for interview. Finally, a total weighted sample of 15,074 children aged 12–59 months was embodied in this study.

### Study variables

The dependent variable was deworming status, which was dichotomized as "poor" and "good". Poor deworming uptake (i.e., a child who had not taken deworming medication), which was labeled as "poor" and coded 1. A child who has taken supplementary deworming medication was said to be good with the deworming drug and labeled as "good" and coded 0.

The independent variables of the study were classified as individual factors and community factors. The individual factors of the variables were maternal age, maternal education, working status, family wealth status, sex of the household head, sex of the child, age of the child, place of delivery, health-related decision-making autonomy, distance to health facilities, and had diarrhea recently, whereas community factors were classified as region and place of residence. The DHS wealth index was categorized into three groups for analysis: poor (poorest + poorer), middle, and rich (richer + richest) [34].

### Statistical analysis

Descriptive statistics were used to summarize the characteristics of the study population. Bivariate associations between deworming uptake and explanatory variables were assessed using Chi-square tests. To identify independent predictors, a multilevel mixed-effects logistic regression model was employed, accounting for clustering at the community level. Data cleaning and analysis were performed using STATA version 17.

### Ethical approval

This study is based on the publicly available secondary dataset, and the initial survey was conducted with proper ethical approval from the respective authorities. Moreover, the initial survey took the proper ethical approval.

## Results

### Sociodemographic characteristics

A total of 15,074 children aged 12–59 months were included in this study. The overall prevalence of poor deworming uptake in Somalia was about 92% (95% CI: 90.33–93.47). Regional variations in poor deworming uptake were observed in Somalia for example Awdal (95%), Woqooyi Galbeed (93.23%) Togdheer (85.15%), Sool (89.52%), Sanaag (88.63%), Bari (90.02%), Nugaal (92.88%), Mudug (93.6%), Galgaduud (94.94%), Hiraan (95.50%), Middle Shabelle (93.87%), Banadir (92.93%), Bay (93.06%), Bakool (93.63%), Gedo (95.93%), and Lower Juba (85.46%). Poor deworming uptake was varied among maternal age groups, with rates of 92.10% (15–24 years), 91.89% (25–34 years), and 91.60% (35–49 years). Poor deworming uptake was higher among the mothers having higher education. It is also observed that poor deworming uptake varies among working and non-working mothers. The variations in poor deworming were observed among family wealth status, poor (96.85%), while 87.06% were observed in rich families. According to variations of poor deworming in perceived distance to health facilities, (91.40%) was not a big problem, while (92.05%) was a big problem. Results depict that 93.37% were at home, while 85.85% were at the health facility. The variations of poor deworming uptake among children of different ages in months were 92.91% between 12–15 months, 91.75% between 16–19 months, and 91.26% between 20–59 months. Of the respondents, 94.95% of rural, 92.99% of urban, and 87.99% of nomadic had poor deworming [Table 1].

**Table 1. Deworming by sociodemographic characteristics.**

| Variables | Categories | Frequency (%) | Deworming | | p-value of χ² |
|---|---|---|---|---|---|
| | | | Good (%) | Poor (%) | |
| Maternal age | 15-24 | 3,964 (26.30) | 313(7.90) | 3,651(92.10) | 0.733 |
| | 25-34 | 7,838 (52.00) | 636(8.11) | 7,202(91.89) | |
| | 35-49 | 3,272 (21.71) | 275(8.40) | 2,997(91.60) | |
| Maternal education | No education | 12,744(84.54) | 901(7.07) | 11,843(92.93) | <0.001 |
| | Primary | 1,763 (11.70) | 237(13.44) | 1,526(86.56) | |
| | Secondary | 439 (2.91) | 58(13.21) | 381(86.79) | |
| | Higher | 128 (0.85) | 28(21.88) | 100(78.13) | |
| Working status | Not Working | 14,950(99.18) | 1,190(7.96) | 13,760(92.04) | <0.001 |
| | Working | 124(0.82) | 34(27.42) | 90(72.58) | |
| Sex of the household head | Male | 10,227(67.85) | 861(8.42) | 9,366(91.58) | 0.051 |
| | Female | 4,847 (32.15) | 363(7.49) | 4,484(92.51) | |
| Family wealth status | Poor | 7,009 (46.50) | 221(3.15) | 6,788(96.85) | <0.001 |
| | Middle | 2,924 (19.40) | 338(11.56) | 2,586(88.44) | |
| | Rich | 5,141 (34.11) | 665(12.94) | 4,476(87.06) | |
| Health-related decision-making autonomy | Mother Alone | 2,688 (17.83) | 197(7.33) | 2,491(92.67) | 0.208 |
| | Husband Only | 7,686 (50.99) | 628(8.17) | 7,058(91.83) | |
| | Jointly | 4,700(31.18) | 399(8.49) | 4,301(91.51) | |
| Perceived distance to health facilities | Not a big problem | 4,012 (26.62) | 345(8.60) | 3,667(91.40) | 0.195 |
| | Big problem | 11,062(73.38) | 879(7.95) | 10,183(92.05) | |
| Place of delivery | Home | 12,092(80.22) | 802(6.63) | 11,290(93.37) | <0.001 |
| | Health facility | 2,982(19.78) | 422(14.15) | 2,560(85.85) | |
| Age of the child(months) | 12–15 months | 1,227 (11.48) | 87(7.09) | 1,140(92.91) | 0.147 |
| | 16–19 months | 570 (5.34) | 47(8.25) | 523(91.75) | |
| | 20–59 months | 8,887 (83.18) | 777(8.74) | 8,110(91.26) | |
| Sex of the child | Male | 7,854 (52.10) | 640(8.15) | 7,214(91.85) | 0.893 |
| | Female | 7,220 (47.90) | 584(8.09) | 6,636(91.91) | |
| Had diarrhea recently | Yes | 794 (5.27) | 261(32.87) | 533(67.13) | <0.001 |
| | No | 14,280(94.730) | 963(6.74) | 13,317(93.26) | |
| Place of residence | Rural | 3,957 (26.25) | 200(5.05) | 3,757(94.95) | <0.001 |
| | Urban | 6,222 (41.28) | 436(7.01) | 5,786(92.99) | |
| | Nomadic | 4,895 (32.47) | 588(12.01) | 4,307(87.99) | |
| Region | Awdal | 640 (4.25) | 32(5.00) | 608(95.00) | <0.001 |
| | Woqooyi Galbeed | 990 (6.57) | 67(6.77) | 923(93.23) | |
| | Togdheer | 963 (6.39) | 143(14.85) | 820(85.15) | |
| | Sool | 1,107(7.34) | 116(10.48) | 991(89.52) | |
| | Sanaag | 1,231(8.17) | 140(11.37) | 1,091(88.63) | |
| | Bari | 882(5.85) | 88(9.98) | 794(90.02) | |
| | Nugaal | 871 (5.78) | 62(7.12) | 809(92.88) | |
| | Mudug | 882 (5.85) | 55(6.24) | 827(93.76) | |
| | Galgaduud | 810 (5.37) | 41(5.06) | 769(94.94) | |
| | Hiraan | 755 (5.01) | 34(4.50) | 721(95.50) | |
| | Middle Shabelle | 799 (5.30) | 49(6.13) | 750(93.87) | |
| | Banadir | 1,783(11.83) | 126(7.07) | 1,657(92.93) | |
| | Bay | 346 (2.30) | 24(6.94) | 322(93.06) | |
| | Bakool | 1,021 (6.77) | 65(6.37) | 956(93.63) | |
| | Gedo | 1,031 (6.84) | 42(4.07) | 989(95.93) | |
| | Lower Juba | 963 (6.39) | 140(14.54) | 823(85.46) | |

## Multilevel logistic regression analysis

Results of the multivariable multilevel logistic analysis presented in Fig 1 and S1 Table show several significant factors associated with poor deworming uptake. Compared to the reference groups, significantly lower odds of poor deworming uptake were observed for mothers aged 35–49 years (AOR: 0.73; 95% CI: 0.59, 0.90, $p<0.05$), mothers with primary (AOR: 0.69; 95% CI: 0.56, 0.84) or higher education (AOR: 0.35; 95% CI: 0.20, 0.60, $p<0.05$), those in the middle (AOR: 0.39; 95% CI: 0.32, 0.49, $p<0.05$) or rich wealth quintiles (AOR: 0.36; 95% CI: 0.29, 0.45, $p<0.05$), and working mothers (AOR: 0.28; 95% CI: 0.16, 0.48, $p<0.05$). Lower odds of poor uptake were also seen when health decisions were made by the husband alone (AOR: 0.74; 95% CI: 0.60, 0.91) or jointly (AOR: 0.76; 95% CI: 0.61, 0.95, $p<0.05$) compared to the mother alone, and for children delivered in a health facility (AOR: 0.59; 95% CI: 0.49, 0.70). Older children aged 20–59 months (AOR: 0.77; 95% CI: 0.60, 0.98, $p<0.05$) also had lower odds of poor uptake compared to those aged 12–15 months. Residing in urban (AOR: 0.65; 95% CI: 0.51, 0.82, $p<0.05$) or nomadic areas (AOR: 0.40; 95% CI: 0.32, 0.49, $p<0.05$) was significantly associated with lower odds of poor uptake compared to rural areas. Conversely, the odds of poor deworming uptake were substantially and significantly higher for children who had no diarrhea recently compared to those who did (AOR: 6.26; 95% CI: 5.11, 7.67, $p<0.05$). Significant variations were also observed across different regions.

## Discussion

This study investigated the prevalence and multilevel determinants of deworming uptake among children aged 12–59 months in Somalia using the nationally representative 2020 Somalia Demographic and Health Survey (SDHS) data. Only 8% of children received deworming medication, highlighting alarmingly poor coverage. This figure starkly underscores the significant public health challenge posed by soil-transmitted helminth (STH) infections in this vulnerable population and falls dramatically short of the World Health Organization's (WHO) goals for preventive chemotherapy (PC) coverage in endemic areas [6,35]. The extremely low coverage highlights the immense gap between recommended interventions and the reality on the ground in Somalia, a context marked by fragility and strained health systems [18–20]. While specific contemporary studies on deworming coverage in Somalia are scarce, as noted in the introduction, the finding of low coverage aligns with challenges documented in other low- and middle-income countries (LMICs), particularly those in Sub-Saharan Africa facing instability or limited resources [9,14]. However, the 8% coverage found here appears exceptionally low, even compared to regional neighbors or other fragile settings where PC programs have gained more traction, suggesting potentially unique or exacerbated barriers within Somalia [15,17]. The significant regional variations observed, with poor uptake ranging from 85% to nearly 96%, mirror findings from other large-scale surveys in diverse settings [31] and underscore the importance of context-specific factors influencing program reach, likely related to varying levels of security, infrastructure, and localized health initiatives [19,20].

The multilevel analysis identified several significant determinants operating at individual, household, and community levels, consistent with findings from studies on child health interventions elsewhere. At the individual and household level, higher maternal age, maternal education (primary and higher), higher family wealth status, maternal employment, and delivery in a health facility were all associated with significantly lower odds of poor deworming uptake (i.e., better coverage). These findings resonate with extensive literature indicating that improved maternal education and socioeconomic status enhance health knowledge, health-seeking behavior, and access to care [4,23,24]. Delivery in a health facility likely represents increased contact with the formal health system, providing opportunities for preventive services like deworming [20]. The protective effect of father or joint decision-making may reflect household dynamics influencing healthcare access. This finding warrants further investigation. Older child age (20–59 months) being associated with better uptake compared to the youngest group (12–15 months) might reflect cumulative opportunities for receiving deworming medication through campaigns or health contacts over time.

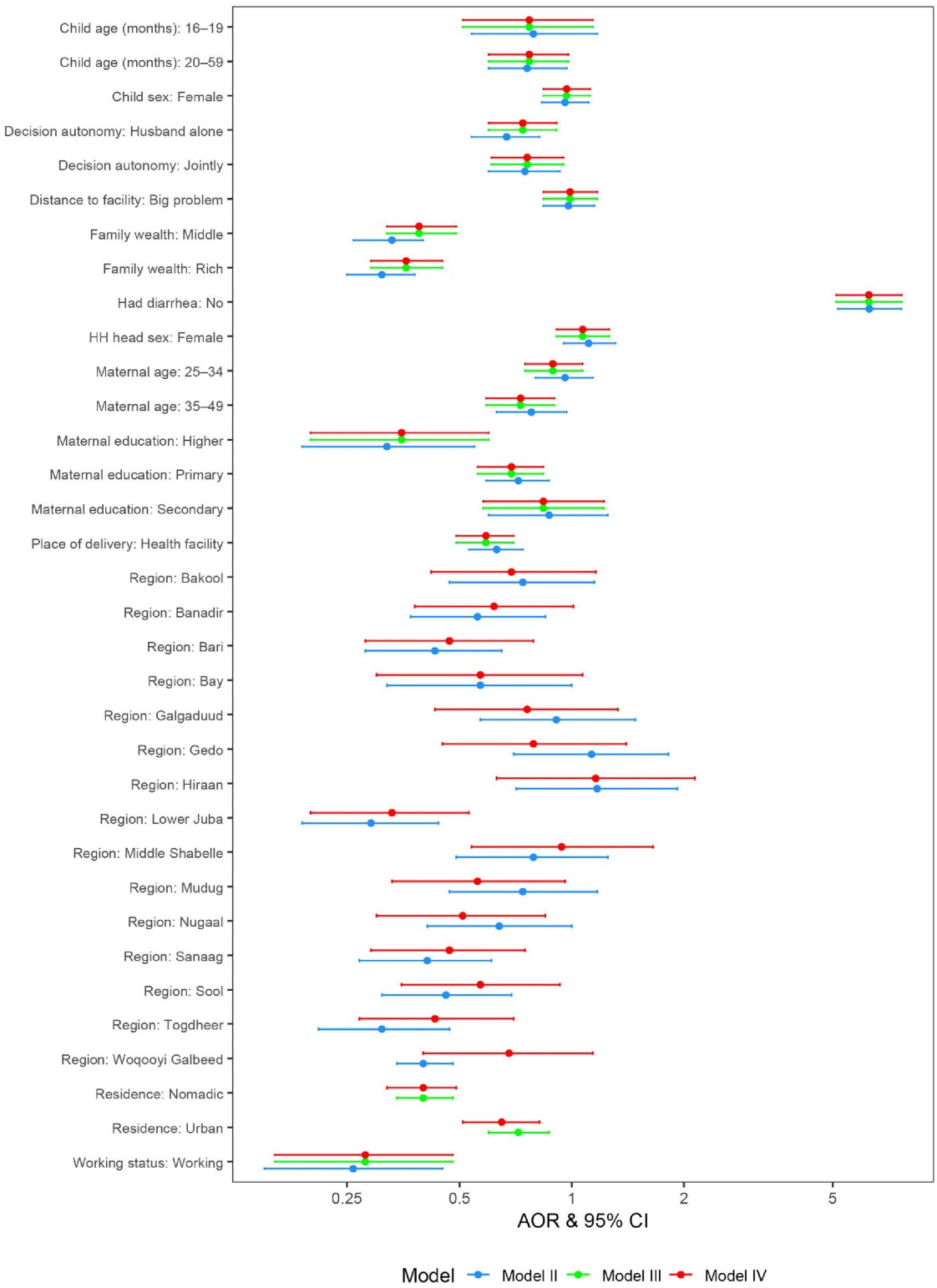

**Fig 1. Forest plot of poor deworming among children of 12–59 months in Somalia.**

At the community level, residing in urban or nomadic areas was protective against poor uptake compared to rural areas. The finding for urban areas aligns with expectations of potentially better access to health facilities and information [25]. The relatively better uptake among nomadic populations compared to rural dwellers is intriguing and could reflect specific outreach programs targeting nomadic groups or perhaps different exposure patterns, although this contrasts with typical assumptions about access challenges for mobile populations [16,22]. The substantial regional differences persisted even after controlling for individual and household factors, highlighting the strong influence of geographic context, which may encompass variations in program implementation, partner presence, local governance, and security [17,19,36]. One of the most striking findings was the strong association between a child not having had diarrhea recently and significantly higher odds of poor deworming uptake. There may be a connection between intestinal parasite infection and diarrhoea, and children who experience diarrhoea are more likely to seek medical attention [25,37]. Children who recently experienced diarrhea were substantially more likely to have received deworming medication [25]. This finding contradicts the expectation that deworming is primarily a preventive measure. It may indicate that deworming medication is often administered therapeutically when children present with diarrheal illness, or perhaps it is co-administered with treatments for diarrhea during health facility visits. Alternatively, caregivers of children with recent diarrhea might be more engaged with health services or more likely to recall any medications given. This requires further exploration to understand if deworming is being missed as a routine preventive measure and primarily used reactively, suggesting that barriers like awareness, drug availability, or cost might be more dominant than simple physical access for preventive care in this context.

This study makes several key contributions. Firstly, it provides a crucial, up-to-date, nationally representative estimate of deworming coverage among under-fives in Somalia, revealing a critical situation. Secondly, by employing a multilevel analytical approach [33], it moves beyond individual-level factors to identify both individual/household and community/regional determinants, offering a more nuanced understanding of the barriers and facilitators within Somalia's complex environment. Thirdly, it highlights specific high-risk groups (e.g., children in rural areas, those from poorer households, those whose mothers have no education) and protective factors (e.g., health facility delivery, maternal education) that can inform targeted interventions. The identification of the strong link between recent diarrhea and deworming uptake is a particularly important finding, warranting programmatic attention.

The findings directly relate to several SDGs [11]. The extremely low deworming coverage poses a significant barrier to achieving SDG 3 (Good Health and Well-being), particularly Target 3.2 (end preventable deaths of newborns and under-fives) and Target 3.3 (end the epidemics of neglected tropical diseases like STH infections). STH infections directly impede progress by causing malnutrition, anemia, and impaired cognitive development [4,5], undermining children's health and future potential. The association of deworming uptake with maternal education and wealth status links the findings to SDG 4 (Quality Education) and SDG 1 (No Poverty), highlighting how broader development factors impact child health outcomes. The disparities linked to place of residence (rural vs. urban/nomadic) and region connect to SDG 10 (Reduced Inequalities). Addressing the underlying drivers of STH transmission, such as poor sanitation and hygiene [2], is also crucial for SDG 6 (Clean Water and Sanitation). Investing in deworming is thus an investment in human capital and is essential for achieving sustainable development goals [10].

## Limitations of the study

This study has limitations inherent to its design and data source. The cross-sectional nature precludes establishing causality between determinants and deworming status. Deworming status was based on maternal recall, potentially introducing recall bias or social desirability bias. The SDHS question might not capture details about the type of drug, dosage, or frequency, and the definition of "poor" (not receiving any) versus "good" (receiving any) is binary and lacks granularity. While multilevel analysis accounts for clustering, unmeasured confounding variables at individual (e.g., specific health beliefs) or community levels (e.g., intensity of local NGO activities, specific security incidents) could still influence the observed associations. The categorization of some variables (e.g., wealth index) involves simplification.

## Conclusion

Significant determinants associated with deworming coverage include higher maternal education, greater wealth, health facility delivery, urban or nomadic residence compared to rural, and older child age. Multifaceted strategies are necessary to increase coverage and work towards SDG 3. These include bolstering routine health service delivery, incorporating deworming into child health contacts, especially during vaccinations and postnatal care, encouraging facility-based deliveries, and putting in place targeted, possibly community-based or campaign-style, deworming programs that specifically target rural populations, low-income households, and mothers with limited educational attainment. Additionally, it is stressed to make clear the connection between treating diarrhoea and administering deworming in order to guarantee suitable preventive measures. In Somalia, addressing these factors is essential to preserving the growth and health of children.

## Supporting information

**S1 Table. Multivariable multilevel analysis result of poor deworming among children of 12–59 months in Somalia.** (DOCX)

**S1 File. Data.** (CSV)

## Author contributions

**Conceptualization:** Abdirahman Omer Ali, Abdisalam Mahdi Hassan.

**Data curation:** Abdirahman Omer Ali, Awo Mohamed Kahie.

**Formal analysis:** Abdirahman Omer Ali, Awo Mohamed Kahie, Suhaib Mohamed Kahie.

**Methodology:** Muhyadin Yusuf Dahir, Md. Moyazzem Hossain.

**Supervision:** Md. Moyazzem Hossain.

**Validation:** Md. Moyazzem Hossain.

**Writing – original draft:** Abdirahman Omer Ali, Awo Mohamed Kahie, Muhyadin Yusuf Dahir, Suhaib Mohamed Kahie, Abdisalam Mahdi Hassan.

**Writing – review & editing:** Md. Moyazzem Hossain.

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
