## [Decision Letter · Decision Letter 0]

14 Sep 2025

Dear Dr. Hossain,

Thank you for submitting your manuscript to PLOS ONE. After careful consideration, we feel that it has merit but does not fully meet PLOS ONE’s publication criteria as it currently stands. Therefore, we invite you to submit a revised version of the manuscript that addresses the points raised during the review process.

We look forward to receiving your revised manuscript.

Kind regards,

Clement Ameh Yaro, Ph.D

Academic Editor

PLOS ONE

Journal Requirements:

Reviewer's Responses to Questions

**Comments to the Author**

1. Is the manuscript technically sound, and do the data support the conclusions?

Reviewer #1: Yes

Reviewer #2: Yes

2. Has the statistical analysis been performed appropriately and rigorously?

Reviewer #1: Yes

Reviewer #2: Yes

3. Have the authors made all data underlying the findings in their manuscript fully available?

Reviewer #1: Yes

Reviewer #2: Yes

4. Is the manuscript presented in an intelligible fashion and written in standard English?

Reviewer #1: Yes

Reviewer #2: Yes

Reviewer #1: Dear editor, thank you very much for giving me a chance to review the paper entitled "Prevalence and determination of deworming among under-five children in Somalia using Multi-level analysis of SDHS 2020 data,"

Manuscript ID: PONE-D-25-36783

General Comments – Minor Revision Required

This manuscript presents a relevant analysis of deworming coverage among children aged 12–59 months in Somalia, using the 2020 SDHS dataset and applying appropriate multilevel modeling. The public health significance is well articulated, and the use of nationally representative data adds robustness. However, the age range includes children under 24 months, who may not be the standard target for deworming in some national programs—this requires clarification or justification. Additionally, while the 2020 data remains valuable, the five-year gap raises questions about current applicability, and should be acknowledged as a limitation. Minor revisions are recommended to address these temporal and age-related considerations, alongside improvements in language, clarity, statistical reporting, and reference formatting.

Specific comments

Title

1. In the title ``Prevalence and determination of deworming among under-five children in Somalia using Multi-level analysis of SDHS 2020 data,`` I suggest to change the word "determination” to "determinants", which is commonly used in public health literature to describe factors influencing an outcome

2. I suggest to change "using Multi-level analysis of SDHS 2020 data," by " : A Multilevel Analysis of the 2020 SDHS Data"

3. The revised title is Prevalence and Determinants of Deworming Among Under-Five Children in Somalia: A Multilevel Analysis of the 2020 SDHS Data

4. 2020 SDHS Data is about 5 years gab needs justifications

Abstract

1. I suggest to change the phrases “infections significantly impair child health in Somalia." by "infections are a major public health concern in Somalia, particularly affecting the health and development of children under five."

2. I suggest to review the objective as "This study aimed to estimate the prevalence of deworming uptake and identify individual- and community-level factors associated with its use among Somali children aged 12–59 months." For clarity that leads to improves precision and aligns with the multilevel analysis approach

3. I suggest to rephrase "This study is based on a secondary dataset having 15,074 children..." to "The analysis used secondary data from the 2020 Somalia Demographic and Health Survey (SDHS), including 15,074 children aged 12–59 months."

4. I recommend to rephrase "poor deworming uptake (not receiving medication)" to: "non-receipt of deworming medication (poor uptake)."

5. I suggest to rephrase "Deworming coverage was critically low, with 92.0% of children exhibiting poor uptake (only 8.0% received medication)." to "Only 8.0% of children had received deworming medication, indicating critically low national coverage."

6. I suggest to rephrase "Counterintuitively, children without recent diarrhea had significantly higher odds of poor uptake (AOR=6.26)." to "Unexpectedly, children without recent episodes of diarrhea had significantly higher odds of not receiving deworming treatment (AOR = 6.26)."

7. Rephrase the sentences of conclusion for clarity.

8. I recommend to organize key words alphabetically for consistency

Introduction

1. I suggest to condense overly long and dense repetitive idea of introduction to improve readability. For example, the consequences of STH infections on under-five children (malnutrition, cognitive impairment, etc.) are mentioned multiple times and can be streamlined.

2. I suggest to break up long and cover multiple idea into shorter, focused paragraphs. For example, one for global context, one for regional/Sub-Saharan Africa, and another for Somalia-specific challenges

3. I recommend to revise the sentence, ``While effective and low-cos treatments exist, ensuring these interventions reach every child in need remains a formidable challenge, especially in resource-constrained and fragile settings`` to ``While effective and affordable treatments exist, delivering them to all eligible children—particularly in fragile settings like Somalia—remains a significant challenge.``

4. I recommend to rewrite ``This study addresses a critical knowledge gap by providing a contemporary analysis of deworming coverage and its determinants among children under five in Somalia, utilizing the most recent nationally to representative dataset – the SDHS 2020. The authors believed that the findings of this study will be to provide evidence-based recommendations to strengthen deworming strategies and improve child health outcomes in the challenging context of Somalia. `` to ``This study is among the first to use multilevel modeling on SDHS 2020 data to assess both individual- and community-level determinants of deworming uptake in Somali children. Unlike traditional analyses, this method accounts for data clustering and context-specific influences, offering more robust and actionable insights. ``

Methods and materials

1. I suggest to correct phrase of subheadings "Sampling Producers" to "Sampling Procedures"

2. I recommend to revise sentence "This study is based on a cross-sectional study design and utilizes secondary data extracted from the Somalia Health and Demographic Surveys (SHDS)-2020."to "This study employed a cross-sectional design using secondary data from the 2020 Somalia Demographic and Health Survey (SDHS). " and the survey is typically abbreviated as SDHS, not SHDS.

3. I suggest to remove redundancy (e.g., “age of the child” is listed twice).and formating long sentences of independent variables and correct upper and lower case of alphabets

4. I suggest to rewrite the sentences as the following and needs to put reference, ``The DHS wealth index was categorized into three groups for analysis: poor (poorest + poorer), middle, and rich (richer + richest). ``

5. I suggest to rewrite statistical analysis as ``Descriptive statistics were used to summarize the characteristics of the study population. Bivariate associations between deworming uptake and explanatory variables were assessed using Chi-square tests. To identify independent predictors, a multilevel mixed-effects logistic regression model was employed, accounting for clustering at the community level. Data cleaning and analysis were performed using STATA version 17. ``

Results

1. I suggest to correct a type, miscalculated CI or misinterpretation error of a sentence in the sentence, "The overall prevalence of poor deworming uptake in Somalia was 92% (95% CI: 10.67–11.99)."Because a 95% CI of 10.67–11.99 doesn't align with a point estimate of 92%.

2. I suggest to rewrite the sentence “Variation of poor deworming among maternal age 15-24(92.10%), 25-34(91.89%) and 35-49(91.60%).” to “Poor deworming uptake was varied among maternal age, with rates of 92.10% (15–24 years), 91.89% (25–34 years), and 91.60% (35–49 years).”

3. I suggest to put "Good" and "Poor" as separate columns with percentages and total N clearly in the table 2 and bold significant AORs or note them in text clearly

4. I recommend to use consistent past tense throughout ("was", not "is") and avoid redundancy and overly wordy phrasing

5. I recommend to add explanations of AIC and BIC under the table

Discussion

1. I recommend to rewrite the sentence “The findings reveal an alarmingly low prevalence of deworming, with only 8% of children reported to have received deworming medication, indicating that 92% had poor uptake.” to “Only 8% of children received deworming medication, highlighting alarmingly poor coverage.” in order to avoid redundancies & wordiness

2. I recommend to rewrite sentence “The protective effect observed when fathers are involved in health decisions (either alone or jointly with the mother, compared to the mother alone) warrants further investigation but may reflect complex intra-household dynamics regarding healthcare access or resource allocation in this specific cultural context.” into “The protective effect of father or joint decision-making may reflect household dynamics influencing healthcare access. This finding warrants further investigation. “to correct overuse of passive voice and Long sentence. “

3. I recommend to add report AORs in Results but not in Discussion as “Higher maternal education (AOR: 0.69 for primary, 0.35 for higher) was strongly associated with lower odds of poor uptake.”

Conclusion

1. I suggest to condense the conclusion as it is long

References

1. I suggest to add DOIs to maintain consistency in the reference number 1 and 2 as an example

2. I suggest to correctly format reference number 26

3. I recommend to merge reference number 6 and 28 as cited twice

Reviewer #2: Ali and co-authors addressed a crucial public health issue by exploring the contributing factors of low prevalence of deworming uptake among Somali children. However, revision is required before accepting this work for possible publication in Plos One. My recommendations are below:

Title

- I think the term ‘prevalence’ is not appropriate here, this is actually ‘coverage’ of deworming uptake. Consistency in terminology should be maintained throughout the manuscript; for example, the term is written as both “multi-level” and “multilevel.” So, the authors can consider the following title: Coverage and determinants of deworming uptake among under-5 children in Somalia: Multilevel analysis of SDHS 2020 data.

Abstract

- “Therefore, this study aimed to assess the prevalence and identify multilevel factors associated with deworming uptake among Somali children aged 12-59 months.” This sentence could be “Therefore, this study aimed to assess the coverage of deworming uptake and associated multilevel factors among Somali children aged 12-59 months.”

- “This study is based on a secondary dataset having 15,074 children aged 12-59 months and extracted from a countrywide cross-sectional survey, the 2020 Somalia Demographic and Health Survey (SDHS). Multilevel logistic regression was used to examine individual (maternal/child characteristics, health service use) and community (residence, region) factors associated with poor deworming uptake (not receiving medication).” No need to mention ‘secondary dataset and countrywide cross-sectional survey’ as people are familiar about the DHS. So, the authors can consider: This study considered/analyzed data of 15,074 children aged 12-59 months from the 2020 Somalia Demographic and Health Survey (SDHS). Chi-square test and multilevel logistic regression were used to examine the associated factors of poor deworming uptake.

- No need to mention “(only 8.0% received medication)”. focus more quantitative findings rather mentioning only factor names.

- ‘Deworming coverage among young children…..’. Be specific and consistent, there is a formal definition of young children, so replace the ‘young children’ with under-5 children.

- ‘Targeted strategies addressing socioeconomic disparities, promoting health facility use, and reaching rural populations are urgently needed.’ What are the targeted strategies authors like to recommend? Recommendation should be very specific rather some broader terms. Why association with diarrhea warrants investigation? It may be due to children with diarrhea visit either health facility or physicians and during that time physicians prescribe deworming medications. I suggest to search literature and discuss it in the discussion section. Remove the sentence: “Strengthening deworming programs is vital for child health and sustainable development in Somalia.”

Keywords

- Under-5 children; Soil-Transmitted Helminths; Deworming; Multilevel analysis; Somalia

Introduction

- Introduction is unnecessarily large, there are some sentences that are not relevant and necessary. So, the introduction should be concise aligning with the objectives. Remove the sentence: ‘The authors believed that the findings of this study will be to provide evidence-based recommendations to strengthen deworming strategies and improve child health outcomes in the challenging context of Somalia.’

Methods and Materials

- Remove “Study Area: Somalia is located in the Horn of Africa, with an estimated surface area of 637,657 km2 and a terrain consisting mainly of plateaus, plains, and highlands. It has the longest coastline in Africa, stretching over 3,333 km along the Gulf of Aden to the north and the Indian Ocean to the east and south. It borders Djibouti along the northwest, Ethiopia to the west, and Kenya to the southwest. Somalia has been described as Africa’s most culturally homogenous country, with around 85% of its residents being ethnic Somalis. The population density in Somalia is about 29 people per square kilometer.”

- The sub-heading ‘Study Design and Data Source’ could be replaced by ‘Study Design, Setting, and Data’

Results

- There are too many results repeated from table. Authors should write only potential findings in the text and avoid decimal point from the text as well as from the table 1.

- Authors can show the results of regression model using forest plot rather table and mention the model performance criteria in the text in order to avoid complexity for readers.

Discussion

- Discussion section needs improvement focusing potential findings.

Conclusion

- Conclusion is too long, it should be very concise.

References

- Among the cited references, most citations are very older. I recommend to cite updated evidences.

**Do you want your identity to be public for this peer review?** For information about this choice, including consent withdrawal, please see our Privacy Policy

Reviewer #1: **Yes: ** Abdulwase Mohammed Seid

Reviewer #2: No

---

## [Author Response · Author response to Decision Letter 1]

22 Sep 2025

Authors' responses to the review comments:

We would like to sincerely thank the anonymous reviewers, and the Academic Editor, for their valuable comments. We have considered all comments and then thoroughly revised and formatted the manuscript. A detailed response to each comment is provided below.

Author's Response to the Editor Comments:

Thank you very much for your comments and feedback. We believe that it helps to enhance the quality of the manuscript. As per comments, a careful revision has been conducted, and all required files are uploaded to the journal submission system. The revised texts are highlighted in “red” color.

Author's Response to the Journal Requirements:

Thank you very much. We revised the manuscript as per PLOS ONE Style.

Thanks. We revised the data availability statement.

Thanks. We revised the manuscript as per your review comments.

Thanks. We reviewed the references and ensured that the list is complete and correct.

Author's Response to the Reviewer 1 Comments:

Thank you very much for your comments and feedback. We believe that it helps to enhance the quality of the manuscript. We revised the manuscript as per your review comments.

The 2020 SDHS is the last country representative data and no new data is available. We check the whole manuscript and fix the grammatical issues and formatting. Revised texts are in “red”.

Thanks for your suggestion. We appetite it. We changed the title as per your suggestion.

The 2020 SDHS is the last countrywide survey data and no new country representative data is available. That’s why we used the 2020 SDHS data. Revised texts are in “red”. Page: 1

Thank you very much for your suggestion. We revised it accordingly. Revised texts are in “red”. Page: 1

Thanks. We revised the sentence. Revised texts are in “red”. Page: 1

Thank you. We revised the manuscript accordingly. The revised texts are in “red”. Page: 1

Thanks. We revised the manuscript as per your recommendation. The revised texts are in “red”. Page: 2

Thank you. We appreciate it. We revised the manuscript as per your suggestion. The revised texts are in “red”. Page; 2

Thanks. We revised the manuscript. The revised texts are in “red”. Page; 2

Thanks. We revised the Conclusion section. The revised texts are in “red”. Page; 2

Thanks. The keywords are arranged in alphabetical order. The revised texts are in “red”. Page: 2

Thank you very much. We delete the repeated texts, The revised texts are in “red”. Page: 2-4

Thanks for your suggestion. We rearranged the Introduction section as per your comments. The revised texts are in “red”. Page: 2-4

Thanks for your recommendation. We appreciate your effort. We revised the manuscript as per your suggestion. The revised texts are in “red”. Page: 3

Thanks. We revised the texts accordingly. The revised texts are in “red”. Page: 4

Thank you very much for your comments. We revised the typo. The revised texts are in “red”. Page: 5

Thank you very much. We revised the manuscript as per your recommendation. The revised texts are in “red”. Page: 5

Thanks. We delete the repetition and correct the typos. The revised texts are in “red”. Page: 5

Thank you. We revised the texts and add a reference. The revised texts are in “red”. Page: 5-6

Thank you very much for your feedback. We revised the texts accordingly. The revised texts are in “red”. Page: 6

Thank you very much for your careful checking of the manuscript and provide the insightful comments. We believe that it helps to enhance the quality and readability of the manuscript. We revised the texts. The revised texts are in “red”. Page: 6

Thanks for your suggestion. We revised the texts accordingly. The revised texts are in “red”. Page: 6

Thanks. We put "Good" and "Poor" as separate columns with percentages in Table 1. We also bold significant AORs for a clear understanding in Table 2. The revised texts are in “red”. Page: 6, 9-10

Thanks. We revised the manuscript. The revised texts are in “red”. Page: 8

Thanks. We add them under Table 2. The revised texts are in “red”. Page: 10

Thank you very much for your suggestion. We revised the manuscript. The revised texts are in “red”. Page: 10

Thanks for your recommendation. We appreciate it. We revised the texts accordingly. The revised texts are in “red”. Page: 10

Thanks. We deleted the AORs from the Discussion section. The revised texts are in “red”. Page: 11-12

Thanks. We revised the Conclusion section. The revised texts are in “red”. Page: 13-14

Thanks. We revised the reference list. The revised texts are in “red”. Page: 15-17

Author's Response to the Reviewer 2 Comments:

Thank you very much for your insightful comments and feedback. We believe that it helps to enhance the quality of the manuscript. We revised the manuscript accordingly. The revised texts are in “red”.

Thanks. We revised the title of the manuscript, The revised texts are in “red”. Page: 1

Thank you. We revised the texts. The revised texts are in “red”. Page: 1

Thanks. We revised the manuscript. The revised texts are in “red”. Page: 1-2

Thank you. We revised the Results section. The revised texts are in “red”. Page: 1-2

Thanks. We revised it accordingly. The revised texts are in “red”. Page: 2

Thanks. We revised it. The revised texts are in “red”. Page: 2

Thanks. We revised the Keywords following the alphabetical order. The revised texts are in “red”. Page: 2

Thanks. The introduction section is revised. The revised texts are in “red”. Page: 2-4

Thanks. We appreciate your comment. We removed it. The revised texts are in “red”. Page: 4-5

Thanks for your suggestion. We revised it. The revised texts are in “red”. Page: 4

Thanks. We rewrite the findings in the Results section. The revised texts are in “red”. Page: 6

Thanks for your suggestion. Actually, showing the results of 3 models in a forest plot presents some difficulties because not all variables were included in the 3 models.

Thanks. We revised the Discussion section. The revised texts are in “red”. Page: 10-12

Thanks. We revised the Conclusion section. The revised texts are in “red”. Page: 13

Thanks. We revised the references.

In conclusion, the revised version of the manuscript has been produced as per the review outcomes. So, we hope that you will be happy to see this greatly improved version. Once again, we would like to thank you all for your dedication, professional services, and cooperation.

---

## [Decision Letter · Decision Letter 1]

15 Oct 2025

Dear Dr. Hossain,

Thank you for submitting your manuscript to PLOS ONE. After careful consideration, we feel that it has merit but does not fully meet PLOS ONE’s publication criteria as it currently stands. Therefore, we invite you to submit a revised version of the manuscript that addresses the points raised during the review process.

We look forward to receiving your revised manuscript.

Kind regards,

Clement Ameh Yaro, Ph.D

Academic Editor

PLOS ONE

Journal Requirements:

Reviewers' comments:

Reviewer's Responses to Questions

**Comments to the Author**

Reviewer #1: (No Response)

Reviewer #2: All comments have been addressed

2. Is the manuscript technically sound, and do the data support the conclusions?

Reviewer #1: Yes

Reviewer #2: Yes

3. Has the statistical analysis been performed appropriately and rigorously?

Reviewer #1: Yes

Reviewer #2: Yes

4. Have the authors made all data underlying the findings in their manuscript fully available?

Reviewer #1: Yes

Reviewer #2: Yes

5. Is the manuscript presented in an intelligible fashion and written in standard English?

Reviewer #1: Yes

Reviewer #2: Yes

Reviewer #1: Dear editor, thank you for the opportunity to review the revised version of the manuscript titled "Coverage and Determinants of Deworming Uptake among Under-Five Children in Somalia: A Multilevel Analysis of the 2020 SDHS Data"

Manuscript ID: PONE-D-25-36783

I have carefully evaluated the authors' responses to my initial comments and appreciate their efforts in addressing most of the concerns raised. However, two important issues remain unresolved:

1. Age Range for Deworming Coverage: The authors continue to report deworming coverage among children aged 12–59 months. However, children aged 12–23 months are not typically included in the target population for deworming in many national programs, where the standard target group is children aged from 24 months. I recommend that the authors either:

Re-analyze or reframe their findings using the 24–59 month age group, in line with national and international deworming guidelines, after confirming the age-specific data in the 2020 SDHS, or

Provide a clear justification—supported by relevant national or international policy or published literature—for including children aged 12–23 months in the analysis to make for readers clear.

2. Inconsistent Confidence Interval Reporting: In the statement “The overall prevalence of poor deworming uptake in Somalia was 92%”, the previously cited 95% confidence interval (10.67–11.99) does not correspond to the reported point estimate and appears to be an error. This discrepancy has not been addressed in the revision. The authors should correct this by reporting an accurate and appropriate confidence interval consistent with the prevalence estimate.

Recommendation: Minor Revision

With these remaining issues adequately addressed, the manuscript would be suitable for publication.

Reviewer #2: The authors have addressed all the comments raised by the reviewers. The authors may consider to use a forest plot instead of table for presenting the results of the regression models.

Best wishes for the authors!

**Do you want your identity to be public for this peer review?** For information about this choice, including consent withdrawal, please see our Privacy Policy

Reviewer #1: **Yes: ** Abdulwase Mohammed Seid

Reviewer #2: No

---

## [Author Response · Author response to Decision Letter 2]

15 Oct 2025

Authors' responses to the review comments:

We would like to sincerely thank the anonymous reviewers and the Academic Editor for their valuable comments. We have considered all comments and then thoroughly revised and formatted the manuscript. A detailed response to each comment is provided in the tables as follows.

Author's Response to Editor Comments:

Thank you very much for your comments and feedback. We believe that it helps to enhance the quality of the manuscript. As per comments, a careful revision has been conducted, and all required files are uploaded to the journal submission system. The revised texts are highlighted in “red” color.

Author's Response to Journal Requirements:

1. Thank you very much. We checked it carefully.

2. Thanks. We checked the reference list and all are OK.

Author's Response to Reviewer 1 Comments:

Thank you very much for your comments and feedback. We believe that it helps to enhance the quality of the manuscript. We revised the manuscript as per your review comments.

We checked and revised our manuscript based on the published literature on children aged 12-59 months. The following. Some of them are given below.

https://doi.org/10.1371/journal.pone.0297377

https://doi.org/10.1371/journal.pntd.0006500

https://doi.org/10.1016/j.jegh.2013.12.005

https://doi.org/10.1155/2023/9529600

https://dhsprogram.com/Data/Guide-to-DHS-Statistics/Micronutrient_Supplementation_and_Deworming_among_Children.htm

Revised texts are in “red”. Page: 4

Thanks for your suggestion. We appreciate it. It was a typo, so we removed it from the manuscript. As per your comments, we have added the corrected 95%CI in the manuscript. Revised texts are in “red”. Page: 6

Author's Response to Reviewer 2 Comments:

Thank you very much for your insightful comments and feedback. We believe that it helps to enhance the quality of the manuscript. We add the forest plot. The revised texts are in “red”. Page: 9

In conclusion, the revised version of the manuscript has been produced as per the review outcomes. So, we hope that you will be happy to see this greatly improved version. Once again, we would like to thank you all for your dedication, professional services, and cooperation.

---

## [Decision Letter · Decision Letter 2]

27 Oct 2025

Coverage and Determinants of Deworming Uptake among Under-Five Children in Somalia: A Multilevel Analysis of the 2020 SDHS Data

PONE-D-25-36783R2

Dear Dr. Hossain,

We’re pleased to inform you that your manuscript has been judged scientifically suitable for publication and will be formally accepted for publication once it meets all outstanding technical requirements.

Kind regards,

Clement Ameh Yaro, Ph.D

Academic Editor

PLOS ONE

Additional Editor Comments (optional):

Reviewers' comments:

Reviewer's Responses to Questions

**Comments to the Author**

Reviewer #1: All comments have been addressed

Reviewer #2: All comments have been addressed

2. Is the manuscript technically sound, and do the data support the conclusions?

Reviewer #1: Yes

Reviewer #2: Yes

3. Has the statistical analysis been performed appropriately and rigorously?

Reviewer #1: Yes

Reviewer #2: Yes

4. Have the authors made all data underlying the findings in their manuscript fully available?

Reviewer #1: Yes

Reviewer #2: Yes

5. Is the manuscript presented in an intelligible fashion and written in standard English?

Reviewer #1: Yes

Reviewer #2: Yes

Reviewer #1: Dear Editor, thank you for the opportunity to re-review the revised version of the manuscript titled “Coverage and Determinants of Deworming Uptake among Under-Five Children in Somalia: A Multilevel Analysis of the 2020 SDHS Data”

Manuscript ID: PONE-D-25-36783R2

I have carefully examined the authors’ revised manuscript and their detailed responses to my previous comments. I am pleased to note that the authors have satisfactorily addressed all the issues raised in the earlier round of review, including clarification of the age range for deworming coverage and correction of the confidence interval reporting. The revisions have substantially improved the clarity, methodological soundness, and overall presentation of the manuscript.

I have no further concerns and find the revised version suitable for publication in PLOS ONE.

Recommendation: Accept

Reviewer #2: Authors revised the manuscript as per the suggestions by the reviewers. My recommendation is to accept this manuscript for possible publication.

**Do you want your identity to be public for this peer review?** For information about this choice, including consent withdrawal, please see our Privacy Policy

Reviewer #1: **Yes: ** Abdulwase Mohammed Seid

Reviewer #2: No

---

## [Editor Report · Acceptance letter]

PONE-D-25-36783R2

PLOS ONE

Dear Dr. Hossain,

I'm pleased to inform you that your manuscript has been deemed suitable for publication in PLOS ONE. Congratulations! Your manuscript is now being handed over to our production team.

Kind regards,

on behalf of

Dr. Clement Ameh Yaro

Academic Editor

PLOS ONE